# The Th2 Response and Alternative Activation of Macrophages Triggered by *Strongyloides venezuelensis* Is Linked to Increased Morbidity and Mortality Due to Cryptococcosis in Mice

**DOI:** 10.3390/jof9100968

**Published:** 2023-09-26

**Authors:** Ludmila Gouveia-Eufrasio, Gustavo José Cota de Freitas, Marliete Carvalho Costa, Eluzia Castro Peres-Emidio, Paulo Henrique Fonseca Carmo, João Gustavo Mendes Rodrigues, Michelle Carvalho de Rezende, Vanessa Fernandes Rodrigues, Camila Bernardo de Brito, Guilherme Silva Miranda, Pâmela Aparecida de Lima, Lívia Mara Vitorino da Silva, Jefferson Bruno Soares Oliveira, Tatiane Alves da Paixão, Daniele da Glória de Souza, Caio Tavares Fagundes, Nalu Teixeira de Aguiar Peres, Deborah Aparecida Negrão-Correa, Daniel Assis Santos

**Affiliations:** 1Departamento de Microbiologia, Laboratório de Micologia, Universidade Federal de Minas Gerais, Belo Horizonte 31270-901, Brazil; milagouveia@yahoo.com.br (L.G.-E.); naluperes@gmail.com (N.T.d.A.P.); 2Departamento de Parasitologia, Laboratório de Esquistossomose, Universidade Federal de Minas Gerais, Belo Horizonte 31270-901, Brazildenegrao91@gmail.com (D.A.N.-C.); 3Departamento de Microbiologia, Laboratório de Interação Microrganismo-Hospedeiro, Universidade Federal de Minas Gerais, Belo Horizonte 31270-901, Brazilsouzadg@gmail.com (D.d.G.d.S.); caio.fagundes@gmail.com (C.T.F.); 4Departamento de Patologia, Laboratório de Patologia Celular e Molecular, Universidade Federal de Minas Gerais, Belo Horizonte 31270-901, Braziltatipaixao.ufmg@gmail.com (T.A.d.P.)

**Keywords:** cryptococcosis, *Strongyloides venezuelensis*, immune response, morbidity, mortality

## Abstract

Cryptococcosis is a systemic mycosis that causes pneumonia and meningoencephalitis. Strongyloidiasis is a chronic gastrointestinal infection caused by parasites of the genus *Strongyloides.* Cryptococcosis and strongyloidiasis affect the lungs and are more prevalent in the same world regions, i.e., Africa and tropical countries such as Brazil. It is undeniable that those coincidences may lead to the occurrence of coinfections. However, there are no studies focused on the interaction between *Cryptococcus* spp. and *Strongyloides* spp. In this work, we aimed to investigate the interaction between *Strongyloides venezuelensis* (Sv) and *Cryptococcus gattii* (Cg) in a murine coinfection model. Murine macrophage exposure to Sv antigens reduced their ability to engulf Cg and produce reactive oxygen species, increasing the ability of fungal growth intracellularly. We then infected mice with both pathogens. Sv infection skewed the host’s response to fungal infection, increasing lethality in a murine coinfection model. In addition to increased NO levels and arginase activity, coinfected mice presented a classic Th2 anti-Sv response: eosinophilia, higher levels of alternate activated macrophages (M2), increased concentrations of CCL24 and IL-4, and lower levels of IL-1β. This milieu favored fungal growth in the lungs with prominent translocation to the brain, increasing the host’s tissue damage. In conclusion, our data shows that primary Sv infection promotes Th2 bias of the pulmonary response to Cg-infection and worsens its pathological outcomes.

## 1. Introduction

Cryptococcosis is a systemic mycosis that affects the lungs and the central nervous system (CNS), causing pneumonia and meningoencephalitis [1]. The main causative agents, i.e., *Cryptococcus neoformans* and *C. gattii*, are environmental fungi found in birds’ feces and trees, which infect hosts by inhalation. This disease causes 152,000 new cases of meningitis annually, resulting in 112,000 deaths [2,3]. One of the factors that contributed to the emergence of cryptococcosis globally was the increased number of HIV-infected patients [4]. Between 1996 and 2006, the frequency of deaths caused by cryptococcosis in AIDS patients was the highest recorded [5].

In addition to HIV, other organisms have been described in coinfections with *Cryptococcus* spp., including *Mycobacterium tuberculosis*, *Klebsiella pneumoniae*, *Trichosporon mycotoxinivorans*, *Pneumocystis jirovecii*, *Talaromyces marneffei*, *Aspergillus*, *Histoplasma capsulatum*, and *Pseudomonas aeruginosa* [6,7,8,9,10,11,12,13]. These coinfections can occur with evolutionarily close organisms and between organisms from different kingdoms and domains, such as Monera, Protista, viruses, and other fungi [6]. In this context, little is known about cryptococcal coinfection with organisms from the Animalia kingdom, such as nematodes [14,15].

Strongyloidiasis is one of the most common diseases caused by nematodes. It is a chronic gastrointestinal infection caused by *Strongyloides* [16,17,18], with *S. stercoralis* being the leading causative agent of human strongyloidiasis [16]. *Strongyloides* larvae infect the host through penetration into the skin and then migrate to the lungs via the bloodstream, causing pneumonia and pulmonary hemorrhage [19]. After that, the larvae migrate to the host’s intestine, where they mature and produce eggs, which hatch in the intestinal lumen, eliminating larvae in the feces [20,21]. Therefore, this parasitosis is associated with a lack of appropriate basic sanitation since the infection occurs because of parasite elimination in human feces in open places [22]. The autoinfection allows larvae to reinfect the host, perpetuating the infection and lung lesions. Severe cases of hyperinfection occur in immunocompromised patients with a high mortality rate (85%). Buonfrate et al. [23] estimated the global prevalence of strongyloidiasis in 2017 to be 8.1%, corresponding to 613.9 million people infected [20,21]. The disease occurs mainly in tropical and subtropical regions and is endemic in Africa, South America, and Southeast Asia [19,24].

Cryptococcosis and strongyloidiasis affect the lungs and are more prevalent in the same world regions, i.e., Africa and tropical countries such as Brazil. It is undeniable that those coincidences may lead to the occurrence of coinfections. However, no studies have focused on the interaction between *Cryptococcus* spp. and *Strongyloides* spp. Understanding how these pathogens interact with the host is of fundamental importance to elucidating the influence of the parasite on fungal disease and vice versa. In this work, we aimed to investigate the interaction between *Strongyloides venezuelensis* (Sv) and *Cryptococcus gattii* (Cg) in a murine coinfection model. In addition, inflammatory mediators and cell recruitment were measured.

## 2. Materials and Methods

### 2.1. Strains

The interaction between Cg and Sv was tested in two different models: an ex vivo macrophage culture and a murine model. For all analyses, *Cryptococcus gattii*—Cg (clinical isolate L27/01 belonging to the culture collection of the Mycology Lab at the Universidade Federal de Minas Gerais) was cultured on Sabouraud Dextrose Agar—SDA (Difco, San Jose, CA, USA) at 37 °C for 48 h [25].

Furthermore, a *Strongyloides venezuelensis* (Sv) isolate was initially obtained from rats (*Rattus norvegicus*) and has been kept for several passages in Wistar rats in the Schistosomiasis Laboratory at the Universidade Federal de Minas Gerais. Infective filiform larvae (L3) were obtained from infected rats’ fecal cultures using the Baermann apparatus, according to Negrao-Correa et al. [26]. The larvae were filtered, washed several times with PBS, and counted in the magnifying glass.

### 2.2. Preparation of S. venezuelensis Antigens

Considering the importance of phagocytosis in the mouse response against Cg, we initially tested the coinfection of Cg–Sv in macrophages. For this purpose, we used a Sv antigen solution obtained from Sv infectious larvae (L3). Briefly, a L3 larvae suspension was resuspended in PBS containing protease inhibitor (Boehringer Mannheim, Indianapolis, USA), disrupted with glass beads (212 to 300 microns, Sigma Chemical, St. Louis, MO, USA), mixed by vortexing, and then the processed larvae were transferred to another tube. Then, the suspension was sonicated using five cycles of 30 s, and the homogenate was centrifuged at 4000× *g* for 30 min. Protein quantification was performed as previously described [27], and the suspension was stored at −20 °C until use [28]. The concentration used in the phagocytosis assay was 10 µg/mL.

### 2.3. Mice

Female BALB/c mice (*n* = 7 mice per group), six to eight weeks old, from the Animal Facility of the Universidade Federal de Minas Gerais, were used. Water, food, and light/dark cycles were provided *ad libitum*. The guidelines of the Brazilian Society of Zootechnics/Brazilian Society of Science in Laboratory Animals (available at https://www.sbcal.org.br; accessed on 1 March 2019) and Federal Law 11794 were followed in this study. In addition, animal studies were approved by the Ethics Committee on Animal Use from the Universidade Federal de Minas Gerais (CEUA/UFMG, protocol No. 369/2018). The animals were anesthetized intraperitoneally with ketamine (80 mg/kg) (Cetamin^®^ Syntec, São Paulo, SP, Brazil) and xylazine (15 mg/kg) (Xilazin^®^ Syntec, São Paulo, SP, Brazil) before experimentation.

### 2.4. Phagocytosis Assay

The interaction between Sv and Cg was initially tested in bone marrow-derived macrophages (BMDM). Bone marrow cells were harvested from the tibias and femurs of six week old female BALB/c mice. Then, cells were cultured in BMDM differentiation medium (RPMI [Gibco], Waltham, MA, USA] supplemented with 30% L929 growth-conditioning media, 20% bovine fetal serum [BFS, Gibco], Waltham, MA, USA], 2 mM glutamine [Sigma-Aldrich, St. Louis, MO, USA], 100 units/mL of penicillin-streptomycin [Gibco, Waltham, MA, USA], and 50 µM of 2-mercaptoethanol [Gibco, Waltham, MA, USA]) for one week at 37 °C/5% CO_2_ with the addition of new media every 48 h [29]. Then, the supernatant was discarded, and the adhered cells were washed with PBS, followed by adding 3 mL of 10 mM PBS/EDTA and incubation on ice for 10 min. Differentiated macrophages were resuspended and transferred to a sterile polypropylene tube, centrifuged at 200× *g*/5 min at 4 °C, and resuspended in 5 mL of RPMI 1640 medium containing 10% BFS, 2 mM glutamine, 25 mM HEPES (Gibco, Waltham, USA), pH 7.2, 100 units/mL G penicillin, and 5% L929 cell culture supernatant. Trypan Blue (Sigma-Aldrich, St. Louis, MO, USA) was used to determine cell viability. Viable cells (5 × 10^4^ cells per well) were plated in 24-well plates for phagocytic index (PI) and intracellular proliferation rate (IPR) determination or in 96-well plates for MTT viability and reactive oxygen species (ROS) and peroxynitrite (PRN) quantification [13,30].

Before Cg infection, macrophages were stimulated with L3 antigens (10 µg/mL) for 3 h. Non-treated cells were used as controls. For the macrophages’ infection, the fungal suspensions of 0.4 × 10^5^ viable cells per well (5:1 yeasts:macrophages) were added to the macrophage culture in 24-well plates containing 13 mm circular coverslips (PERFECTA, São Paulo, SP, Brazil). Cultures were incubated at 37 °C in 5% CO_2_ for 3 and 24 h after infection with *C. gattii* [31].

After incubation (for 3 h or 24 h), the coverslips were carefully removed, washed in sterile PBS, dried, fixed with ice-cold methanol, and stained with Panotico Rapido (LABORCLIN, Pinhais, PR, Brazil). Macrophage counting was performed by optical microscopy, and the percentage of internalized yeasts expressed the phagocytic index.

To investigate the number of viable yeasts inside these macrophages, an assay was performed as previously described [31]. Briefly, non-internalized yeasts from the supernatant were removed by washing the wells with 500 µL of PBS. Then, macrophages were lysed after 3 h or 24 h with 200 µL of sterile water and incubated for 30 min at 37 °C. After that, 50 µL of the lysate was collected and plated on SDA (Difco, San Jose, CA, USA). The colonies were counted, and the results were expressed as counting forming units (CFU) per mL.

To quantify reactive oxygen species (ROS) and peroxynitrite (PRN), 2,7-dichlorofluorescein diacetate (DCFH-DA; Invitrogen, Life Technologies, Carlsbad, CA, USA) and dihydrorhodamine 123 (DHR 123; Invitrogen, Life Technologies) were used, respectively. After incubation for 30 min at 37 °C, the fluorescence was read at excitation wavelengths of 485 nm and emission wavelengths of 530 nm. The data were expressed as arbitrary fluorescence units ± SE [32]. To evaluate if exposure to the L3 antigen affected the viability of macrophages, the 3-(4,5-dimethylthiazol-2-yl)-2,5-diphenyltetrazolium bromide (MTT) colorimetric assay was performed [13].

### 2.5. Coinfection Protocol: Mouse Survival and Behavior

To evaluate the influence of Sv during mouse Cg infection, we performed coinfection experiments using two distinct protocols: (i) Sv infection seven days before and seven days after Cg, and (ii) Sv infection two days before and two days after Cg. The inoculum of 1 × 10^4^ CFU/30 µL of Cg was used to infect mice intratracheally under anesthesia [25]. A total of 700 L3 of Sv in 100 µL of PBS were subcutaneously inoculated in mice [28]. Groups of non-infected (NI) and mono-infected animals (only with Cg or Sv) were included as controls. The animals were monitored daily for survival, weight, and behavior.

Mouse behavior was assessed using the SHIRPA (SmithKline/Harwell/Imperial College/Royal Hospital/PhenotypeAssessment) protocol [33]. This protocol provides reliable information regarding murine cerebral dysfunction and its general status. Individual parameters evaluated were grouped into five functional categories: neuropsychiatric state, motor behavior, autonomic function, muscle tone and strength, and reflex and sensory function. Individual parameters were summed up to determine a total score for each category.

#### 2.5.1. Fungal Burden, Differential Leukocyte Counting, and Histopathology

Based on the survival and behavior results, we chose the timepoint of Sv two days before Cg infection for the follow-up experiments. Then, other groups of mice were infected following this coinfection protocol, and mice were euthanized under anesthesia to obtain the bronchoalveolar lavage fluid (BALF), lungs, and brain after ten days of Cg infection. The organs and BALF were aseptically removed, and the lungs and brains were weighed and ground in sterile PBS. Then, they were plated on SDA and incubated for 48 h at 37 °C. The colonies were counted, and the results were expressed in CFU/g or CFU/mL. In addition, differential leukocyte counting was also performed in BALF using Panotico Rapido staining (LABORCLIN, Pinhais, PR, Brazil). Histological evaluation was performed in an independent experiment to obtain animal lungs ten days after Cg infection. Lungs were fixed in formalin, embedded in paraffin for histological sections (5 µm), stained with Hematoxicillin–Eosin (HE), and followed by microscopic observation. Inflammatory parameters were evaluated: inflammation associated with the fungus, perivascular inflammation, interstitial thickening, and amount of fungus. These parameters were classified with scores from 0 (absent) to 3 (present and accentuated).

#### 2.5.2. Inflammatory Response of Coinfected Mice

Considering the reduced survival of mice previously infected with Sv, we performed experiments to characterize the mouse immune response during coinfection. Fragments of lung tissue (100 mg) were homogenized with 1 mL of PBS containing protease inhibitors (0.1 mM phenylmethylsulfonyl fluoride, 0.1 mM benzethonium chloride, 10 mM EDTA, and 20 KI aprotinin A, all purchased from Sigma-Aldrich) and 0.05% Tween 20. Initially, we determined the activity of myeloperoxidase (MPO, an indirect measurement of neutrophil influx) [34], N-acetyl-glucosaminidase (NAG, an indirect measure of macrophage content) [35,36], eosinophil peroxidase (EPO, an indirect measurement of eosinophil influx) [37], and arginase [38,39] in the lungs. Nitric oxide levels were also quantified according to the Griess protocol [40]. Furthermore, IL-4, IL-5, IL-10, IL-17, IL-1β, TNF-α, IFN-γ, TGF-β, CXCL2, and CCL24 were measured by ELISA using commercially available antibodies from DuoSet Kits (R&D Systems, Minneapolis, MN, USA) according to the manufacturer’s instructions.

In addition to the inflammatory mediators, cell influx into the lungs was analyzed by flow cytometry. Lungs were harvested and processed for leukocyte isolation as previously described [41]. Cell suspensions were counted in trypan blue dye, and one million cells were incubated with Fc-block (BD-Biosciences, San Jose, CA, USA) and stained with a mixture of fluorochrome-conjugated antibodies purchased from BioLegend, San Diego, CA, USA (anti-Gr-1-PE; anti-CD45-PerCP/Cy5.5; anti-F4/80-PE/Cy7; anti-CD206-APC; anti-Siglec-F-BV421; anti-CD11c-BV510). After washing and permeabilization, cells were stained with FITC-conjugated anti-Arginase-1 antibody (BioLegend, San Diego, CA, USA). Then, cells were suspended in PBS and acquired in a BD FACSCanto II flow cytometer (BD-Biosciences, San Jose, CA, USA) using BD FACSDiva software. Analyses were performed using the FlowJo software (Tree Star, Ashland, OR, USA). Cell populations were identified using a sequential gating strategy, and the percentage of cells in the live/singlets gate was multiplied by the number of live cells (after trypan blue exclusion) to obtain an absolute live-cell count.

### 2.6. Statistical Analysis

The results were analyzed using statistical tests to compare the significant differences between the groups, with the instrumental support of the PRISMA 6.0 software (GrapPad Inc., San Diego, CA, USA). The 95% significance level was considered so that the values are significantly different (*p* < 0.05). The ANOVA test was applied for the phagocytosis, IPR, ROS, and PRN assays, followed by a Tukey test to compare different groups. The log rank test was used for evaluating the survival rate of animals, and ANOVA followed by Tukey’s multiple comparison tests were used for the behavior and flow cytometry analysis. For CFU/g, the T-test/Mann–Whitney test was applied. A two-way ANOVA followed by Dunett’s multiple comparison tests was used to analyze BALF’s differential count. One-way ANOVA test/Newman–Keuls multiple comparison tests were used for the cytokine analysis and chemokine levels and for the enzymatic assays.

## 3. Results

### 3.1. Sv Impairs Cg Phagocytosis, Alters Macrophage’s Oxidative Response, and Increases Fungal Intracellular Proliferation Rate

Considering the critical role of phagocytosis in controlling cryptococcosis, we evaluated the effect of Sv antigens on the response of macrophages to the infection with *C. gattii*. Cell viability was not affected by exposure to the L3 antigens (Appendix A). As shown in Figure 1A, Sv antigens significantly reduced the Cg phagocytic index 24 h after infection. Despite the reduced phagocytosis, engulfed yeasts were still viable inside the Sv-stimulated macrophages (Figure 1B). Since oxidative stress plays a vital role in intracellular fungal killing, we quantified the levels of ROS and PRN during macrophage infection. Interestingly, the Sv stimulus reduced the levels of ROS (Figure 1C) at all times of Cg infection. However, PRN increased at 24 h (Figure 1D). These data suggest that previous exposure to Sv interferes with macrophage responses to Cg infection.

### 3.2. Sv Infection Increases Morbidity and Mortality in Cg-Infected Mice

To choose the best time for the Sv and Cg coinfection, animals were infected with 10^4^ viable yeast cells and 700 L3 nematode larvae, as previously standardized [13,28], in two coinfection protocols (Figure 2). In the first protocol (Figure 2A–G), animals were infected with L3 seven days before (Sv + Cg) or 7 days after (Cg + Sv) Cg infection. We considered the day of infection with Cg as day 0. For each time of coinfection, a group infected only with the nematode was used as a control (Sv1—before, and Sv2—after day 0). Throughout the analyzed period, animals infected only with *S. venezuelensis* (Sv1 and Sv2) were alive, similar to the non-infected group (NI) (Figure 2A), and showed no weight loss (Figure 2B) or behavioral alteration (Figure 2C–G). On the other hand, animals infected with the nematode 7 days after fungal infection (Cg + Sv) showed reduced survival compared to animals infected with the fungus alone (Figure 2A). Mice infected only with Cg had a mean survival of 27 days, while the group that received the parasite seven days after the fungus (Cg + Sv) had a mean survival of 22 days (*p* < 0.002) (Figure 2A). This coinfected group also presented weight loss and reduced scores in three functional categories of the behavioral study protocol: muscle tone and strength, motor behavior, and autonomous function (Figure 2C,D,F). The group infected with Sv before Cg (Sv + Cg) had a mean survival of 25 days (Figure 2A), weight loss, and behavior with no significant differences from the Cg group (Figure 2C–G).

Then, we evaluated the survival and behavior of mice infected with Cg in a two day difference from Sv infection (Figure 2H–N). This time point relates to the Sv larvae achieving the lungs after two days of infection. Thus, animals were infected with Sv two days before (Sv + Cg) or 2 days after (Cg + Sv) Cg infection. The animals infected only with Cg had a mean survival of 24 days, while the group that received the nematode two days after the fungus (Cg + Sv) had a mean survival of 22 days (*p* < 0.0001) (Figure 2H). This coinfection also led to mice’s weight loss (Figure 2I) and altered motor behavior (Figure 2K). Otherwise, the group infected with the nematode before Cg (Sv + Cg) also showed reduced survival (*p* = 0.0011) compared to the Cg monoinfected group (Figure 2H), along with weight loss (Figure 2I).

Considering the reduced survival of mice infected with Sv two days before Cg and the fact that these two pathogens inhabit the lungs at this time, this protocol was selected to analyze the pathogenesis of this coinfection further.

### 3.3. Previous Infections with Sv Increase the Fungal Burden in the Organs

According to the two day described protocol, animals infected with Cg, Sv, and Sv + Cg were euthanized under anesthesia 10 days after Cg infection to obtain organs and BALF. Interestingly, mice from the Sv + Cg group presented increased fungal burden in the BALF (Figure 3A) and lungs (Figure 3B). Although no significant difference was found for brain CFU, 42% of the monoinfected mice presented Cg in the brain. Fungal burden was detected in 71% of the coinfected mice (Figure 3C).

In addition to analyzing fungal load, cell recruitment in BALF was evaluated. Mononuclear cells were predominant in the non-infected (NI) and infected only with Sv groups. In contrast, in the groups infected only with the fungus (Cg) and that received the parasite before the fungus (Sv + Cg), polymorphonuclear cells were predominant (Figure 3D). Additionally, there was an increased percentage of mononuclear cells in the Sv + Cg group compared to Cg-monoinfected mice (Figure 3E).

The lungs’ histopathological analysis showed the same cell recruitment profile observed in BALF. In the Sv-monoinfected mice, there were multifocal areas of interstitial thickening with a slight increase in cellularity composed of neutrophilic and lymphohistiocytic infiltrate (Figure 3G). In the Cg-monoinfected group (Figure 3H), there was a perivascular inflammatory infiltrate (score 2) and interstitial thickening (score 3) composed predominantly of intact and degenerated neutrophils and a moderate to intense number of foamy macrophages. In addition, a moderate amount of fungus was observed (score 2). In the Sv + Cg group (Figure 3I), there was a perivascular inflammatory infiltrate (score 2) composed predominantly of macrophages and lymphocytes, foamy macrophages, and rare neutrophils, in addition to interstitial thickening (score 2). In this group, many fungi were observed in the alveolar lumen and bronchioles (score 3). Histopathological analysis was performed on the animals’ brains at the same time of infection. However, no changes were observed in the evaluated groups.

### 3.4. Sv–Cg Coinfection Increases Recruitment of Eosinophils and CD206^+^ Arginase-1^+^ Macrophages and Production of IL-4 in the Lungs

We performed experiments to characterize the inflammatory response and cellular recruitment in the lungs of Sv and Cg-coinfected mice. Interestingly, mice from the Sv + Cg group presented increased arginase activity and NO production, together with higher activity of EPO (an indirect measurement of the eosinophil influx to the lungs) (Figure 4A–C). Otherwise, MPO and NAG activities were similar for the Cg and Sv + Cg groups (Figure 4D,E).

Flow cytometry analysis was carried out using lung cells from the Sv, Cg, and Sv + Cg groups (Figure 5). Despite no differences in total cell count and neutrophil numbers, Sv induced a heightened influx of eosinophils and macrophages to the lungs of coinfected mice, as attested by the subpopulations expressing F4/80 and CD11c markers. In addition, macrophages from coinfected mice presented higher expression of arginase as well as CD206^+^, which are markers for alternative macrophage polarization.

Moreover, there was a higher concentration of IL-4 and CCL24 and a lower concentration of IL-1β in the lung tissue of the Sv + Cg group compared to the Cg group. No differences among the analyzed groups were observed in IL-5, IFN-γ, and TGF-β concentrations (Figure 6).

## 4. Discussion

Different studies on coinfections with *Cryptococcus* spp. and other microorganisms have already been carried out [13,42]. These studies highlighted the importance of coinfections in the course and outcome of cryptococcosis. However, no studies have been reported involving this fungus and nematode infections. Considering that diseases caused by nematodes, such as *Strongyloides stercoralis*, are prevalent in the same regions where cryptococcosis has a high occurrence [2,19,24], we hypothesized that coinfections between *Cryptococcus* spp. and *Strongyloides* are plausible. From our point of view, using murine models to study coinfections provides controlled and reproducible protocols. It may help to understand and thus prevent the complications observed in cryptococcosis patients. Specifically regarding strongyloidiasis, *S. stercoralis* cannot cause murine infections [16], justifying the use of Sv in our study.

Our first approach was to study the influence of Sv antigens on Cg phagocytosis and killing by macrophages. These cells are involved in the innate and adaptive host response to infections, including Sv [43]. A previous study demonstrated that IL-4-stimulated macrophages (alternatively activated macrophages) efficiently kill *S. stercoralis* larvae ex vivo [44]. Otherwise, alternatively activated macrophages skewed under the influence of Th-2 cells secreting IL-4 have impaired antifungal activity [45]. In the infection caused by *C. neoformans*, the increased production of the type 2 cytokine IL-13 increased lung fungal burden and reduced mouse survival. This cytokine induces the formation of M2 macrophages with the expression of arginase 1, mannose receptor, and YM1 protein [46]. If M2 macrophage activation is detrimental to the host during fungal infection, classical macrophage activation plays an essential role in infection control. Hardison et al. [47] conducted a study to evaluate the role of Interferon-γ in the progression of cryptococcosis using a transgenic strain that produces murine Interferon-γ, *C. neoformans* H99γ. In that study, animals infected with this strain had a reduction in fungal load and increased production of the Th1-type cytokine IL-17, resulting in a polarization of M1 macrophages. During Cg infections, alveolar macrophages engulf the pathogen and modulate the adaptive immune response [48]. However, Cg developed mechanisms to prevent macrophage activity, such as increasing cell size (impairing fungal engulfment) and polysaccharide capsule production, allowing intracellular survival and multiplication [49,50]. In our results, macrophage exposure to L3 antigens altered the response to Cg infection, characterized by a reduced ability to engulf the fungus after 24 h of infection. In addition, macrophages’ fungicidal activity was impaired, as evidenced by the increased Cg intracellular survival and proliferation. We hypothesized that this low efficiency in fighting the fungus may be due to the reduced levels of ROS production during infection. These results point out that a previous infection with Sv may impair the phagocytosis and fungicidal activity of macrophages, which may lead to a worsening during the disease progression.

These data encouraged us to conduct mouse experimentation with both pathogens. Interestingly, the Sv infection increased mouse morbidity (evidenced by behavior alteration) and mortality in the animals. When Sv infection occurs after Cg, regardless of the time (2 or 7 days), there is an increase in morbidity and mortality in the animals, indicating that the nematode promotes a worsening of their condition, probably by triggering an immune response that impairs the protective mechanisms against the fungus. However, when the nematode is inoculated before Cg, the disease worsens only when the nematode infection occurs two days before Cg. These results attracted our attention because this time point coincides with the passage of Sv larvae through the mice’s lungs during experimental strongyloidiasis [24,51,52]. Thus, both pathogens are inhabiting the same host tissue, at least temporarily, which made us choose this protocol for the subsequent experiments. Moreover, *Strongyloides stercoralis* infection in humans is capable of producing a persistent level of autoinfection, leading to the constant migration of parasite larvae through the host lungs [20], which would be relevant for the outcome of cryptococcosis in possible coinfected individuals.

Further, we recovered higher levels of fungal colonies from the alveolar space (expressed as the CFU in the BALF) and the lung parenchyma of mice previously infected with Sv. It points to the increased host permissibility of fungal multiplication. Regarding CNS colonization by Cg, previous Sv infection predisposes fungal translocation from the lungs to the brain. It is possible that an increased fungal load in the lungs and a decreased macrophage’s fungicidal activity provided a suitable environment for fungal translocation through the Trojan horse mechanism [53]. Further characterization of the host’s inflammatory response revealed that BALF from Cg-monoinfected mice presented a predominance of polymorphonucleated cells, with neutrophils being the main cell type found [54]. On the other hand, the Sv + Cg group exhibited a higher proportion of mononucleated cells, as demonstrated by counting cells in BALF and histopathology.

Considering that the Th1 response is associated with a better prognosis in individuals infected with *Cryptococcus* spp. and that the Th2 response, which is predominantly induced by the nematode infection, is more harmful to the host, cytokine and enzymatic markers of cellular activation were measured in the lung tissue of experimental animals. Oliveira-Brito et al. [55] evaluated the role of iNOS/Arginase-1 in the infection caused by *C. gattii*. They observed that the expression of iNOS and arginase increased during the fungal infection in a murine model. In addition, mice deficient in the expression of iNOS had a more prolonged survival than wild-type animals, despite the higher fungal load in the lungs of the knockout animals than in the wild-type animals [55]. In our study, the Sv + Cg group showed a higher production of NO and arginase, which may have contributed to reduced survival and increased fungal burden during coinfection. Chiapello et al. [56] pointed out that NO can act in the mediation of the apoptosis of inflammatory cells. On the other hand, NO modulation appears to be a determinant factor in strongyloidiasis since NO synthase inhibition increases fecal eggs and larvae in the lungs [57].

It is well established that the migration of *S. venezuelensis* larvae through the lungs of rats or mice induces a type-2 immune response with a local eosinophilic inflammation process and increased IgE concentrations [24,58]. On the other hand, some reports have demonstrated cases of eosinophilia both in immunosuppressed patients and healthy individuals affected by cryptococcosis [59,60,61]. Furthermore, Gao et al. [62] observed high levels of eosinophils and IgE during disseminated cryptococcosis in children, but these levels decreased after antifungal therapy. During nematode infection, there is an increase in IL-5 secretion, which acts on the recruitment and activation of eosinophils [24,63]. There was no difference in IL-5 levels comparing the Sv + Cg and the Cg groups. Still, there was an increased eosinophil recruitment during coinfection, which may be due to the high level of the chemokine CCL24, among other factors [64]. One hypothesis for this result is that IL-5 levels could have been reduced after the recruitment and activation of eosinophils, reaching a level similar to that observed for the group infected only with the fungus.

Aside from the significant increase in CCL24 and eosinophil recruitment, the lung homogenate of Sv + Cg-coinfected mice also had a significantly higher concentration of IL-4 and lower IL-1β than the Cg-infected group. CD4 + T lymphocytes, mast cells, and eosinophils mainly produce IL-4. Among the functions of this cytokine are the induction of IgE production and activation of M2 macrophages, which have been associated with the type-2 immune response induced by Sv infection [24,43,58]. The IL-4-mediated Th2 response triggered by Sv exacerbated cryptococcosis and increased lethality since a lower production of proinflammatory cytokines is associated with a worsening of the fungal disease [55,65]. This phenotype may also result from lower levels of the proinflammatory cytokine IL-1β, which is important for the mouse response against *Cryptococcus* [13,66]. Considering that Th1 and Th17 cells are associated with an adequate antifungal response, Sv may have switched the mouse response to Th2, worsening the disease in the Sv + Cg group. Hence, eosinophils have been shown to produce IL-4 in the lungs of *C. neoformans*-infected mice, and eosinophil-deficient mice presented enhanced Th1 and Th17 responses [67].

Our data shows that primary Sv infection promotes a stronger Th2 bias in the pulmonary response to Cg. While Th2 has already been demonstrated to aid the host response against the parasite [26,39,68,69], it most likely contributed to undesirable M2 macrophage polarization in the lungs, fungal expansion, reduced phagocytosis, increased fungal intracellular proliferation, and increased inflammatory lung pathology, as well as CNS dissemination.

In conclusion, coinfection with Sv and Cg resulted in higher morbidity and mortality in the animals, presumably via Th2-skewing the host’s response to the fungal infection.

## Figures and Tables

**Figure 1 jof-09-00968-f001:**
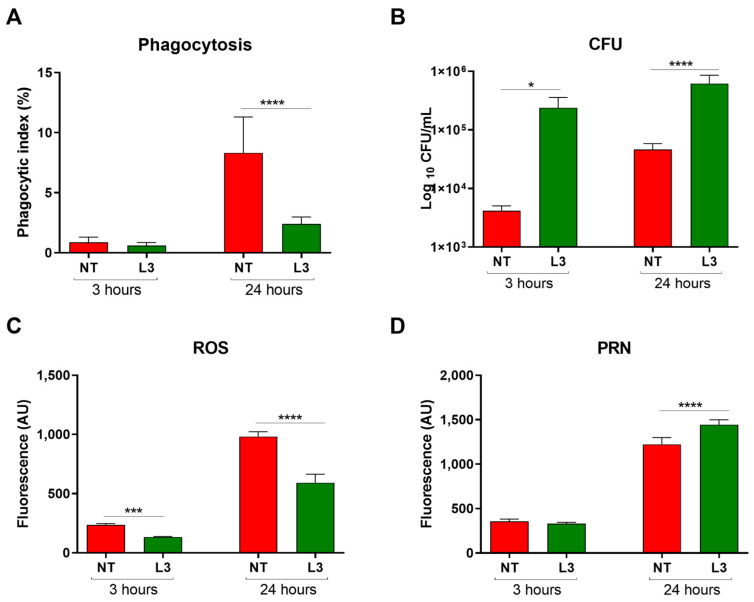
Effect of Sv antigens on in vitro *C. gattii* phagocytosis by macrophages derived from mouse bone marrow. Macrophages were stimulated with L3 antigens for 3 h and incubated with *C. gattii* for 3 or 24 h. (**A**) The phagocytic index, after 3 and 24 h of incubation, is represented by the percentage of yeasts internalized by 300 macrophages. (**B**) Intracellular proliferation and yeast viability after 3 and 24 h of incubation. (**C**) Quantification of ROS after 3 and 24 h of incubation in the presence of *C. gattii*. (**D**) Quantification of PRN after 3 and 24 h of incubation in the presence of *C. gattii*. NT: control group, not stimulated with L3 and infected with Cg; L3: macrophages exposed to L3 antigens for 3 h before incubation with *C. gattii*. ANOVA test/Tukey’s multiple comparisons test: * *p* < 0.05, *** *p* < 0.0002, and **** *p* < 0.0001, representing a statistical difference of phagocytic index, fungal load, and ROS/PRN compared to the NT group. The data are representative of three independent experiments consisting of six replicates each.

**Figure 2 jof-09-00968-f002:**
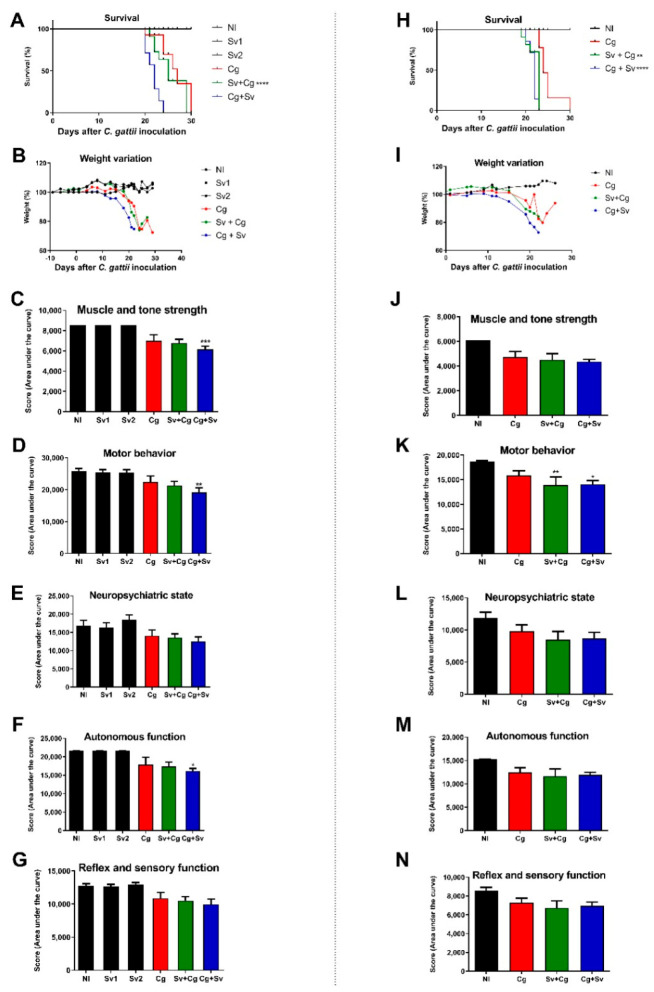
Survival and morbidity of mice infected with Cg or coinfected with Sv in the 7 day (**A**–**G**) and 2 day (**H**–**N**) protocols. (**A**) Survival curve of mice infected intratracheally with 10^4^ viable Cg cells and/or 700 L3 Sv larvae subcutaneously 7 days before (Sv + Cg) or 7 days after (Cg + Sv) Cg inoculation. **** *p* < 0.002 represents the statistical difference compared to the Cg group using the log-rank test. (**B**) Weight variation of the animals, expressed as a percentage. (**C**) Muscle tone and strength. (**D**) Motor behavior. (**E**) Neuropsychiatric status. (**F**) Autonomous function (**G**) Sensory function and reflex. (**H**) Survival curve of mice infected intratracheally with 10^4^ viable Cg cells and/or 700 L3 Sv larvae subcutaneously 2 days before or 2 days after Cg inoculation. ** *p* = 0.0011 and **** *p* < 0.0001 represent a statistical difference in relation to the Cg group using the log-rank test. (**I**) Weight variation of the animals, expressed as a percentage. (**J**) Muscle tone and strength. (**K**) Motor behavior. (**L**) Neuropsychiatric state. (**M**) Autonomous function. (**N**) Sensory function and reflex. ANOVA followed by Tukey’s multiple comparison test: * *p* < 0.5, ** *p* < 0.05, and *** *p* < 0.005 represent statistical differences compared to the Cg group. NI: non-infected; Sv1 and Sv2: groups infected only with *S. venezuelensis*; Cg: groups infected only with *C. gattii*.

**Figure 3 jof-09-00968-f003:**
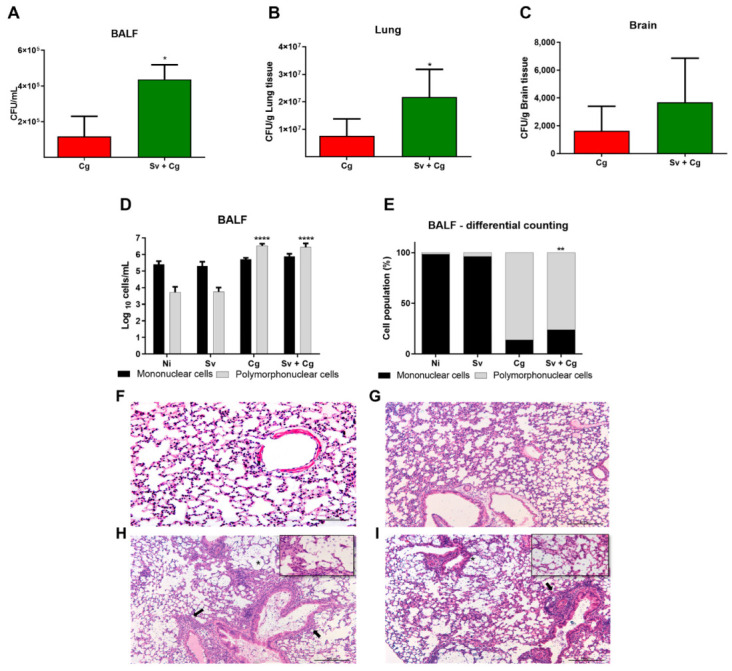
Effect of Sv-Cg mouse coinfection on the fungal burden in the bronchoalveolar lavage fluid (BALF) (**A**), lung (**B**), and brain (**C**). Mice were infected with 700 Sv infective larvae subcutaneously and 2 days later coinfected with 10^4^ viable Cg yeasts, intratracheally, and euthanized 10 days after inoculation of Cg. *T*-test/Mann–Whitney test: * *p* < 0.03 represents a statistical difference compared to the Cg group. Differential cell count in BALF. (**D**) BALF cell recruitment profile, in absolute values—BAL cells/mL, 10 days after Cg inoculation. Two-way ANOVA followed by Dunett’s multiple comparison test: **** *p* < 0.001 shows a statistical difference compared to the NI group. (**E**) BALF cell recruitment profile, differential count—percentage values of mononuclear and polymorphonuclear cells. Two-way ANOVA followed by Dunett’s multiple comparison test: ** *p* < 0.01 shows a statistical difference compared to the Cg group. (**F**–**I**) Histopathological HE staining of the lungs. Representative pictures of the histopathology after ten days of Cg infection. (**F**) NI group: No structures compatible with fungus were seen. (**G**) Sv group: multifocal areas of interstitial thickening with a slight increase in cellularity composed of neutrophilic and lymphohistiocytic infiltrates. No structures compatible with fungus and perivascular inflammation were visualized. (**H**) Cg group: perivascular and interstitial inflammatory infiltrate, predominantly composed of neutrophils, moderately multifocal (arrows). In detail, fungi compatible with *Cryptococcus* (asterisk) can be observed in the alveolar lumen and bronchioles, forming nests and areas of dilation and rupture of the alveolus wall. (**I**) Sv + Cg group: lymphohistioplasmocytic perivascular inflammatory infiltrate (arrows). In detail, alveolar lumen and bronchioles with foamy macrophages and an intense amount of fungus compatible with *Cryptococcus* (asterisk) form nests and areas of dilation and rupture of alveoli. Scale bars are 200 µm. All the experiments were performed at least twice to confirm the data, and the results were always reproducible.

**Figure 4 jof-09-00968-f004:**
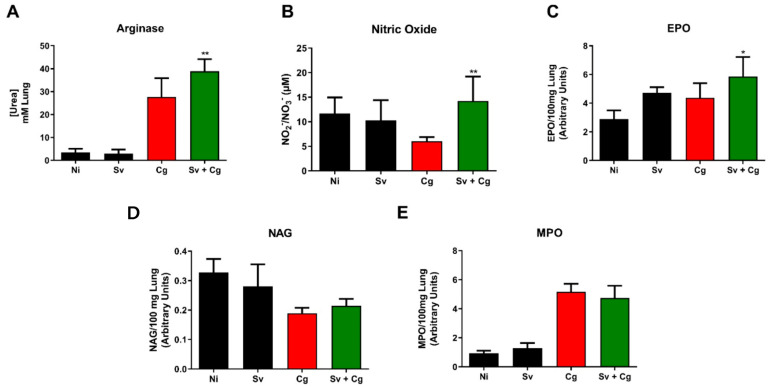
Effect of Sv-Cg coinfection on cell recruitment to the lungs. Arginase activity (**A**), nitric oxide (**B**), and relative numbers of eosinophils (**C**), macrophages (**D**), and neutrophils (**E**) were determined in the lungs of infected mice. One-way ANOVA test/Newman–Keuls multiple comparison test: * *p* < 0.05 and ** *p* < 0.001 represent statistical differences compared to the Cg group. NI: non-infected; Cg: group infected only with *C. gattii*; Sv: group infected only with *S. venezuelensis*; Sv + Cg: group infected with Sv 2 days before infection with *C. gattii*. The data are representative of two independent experiments.

**Figure 5 jof-09-00968-f005:**
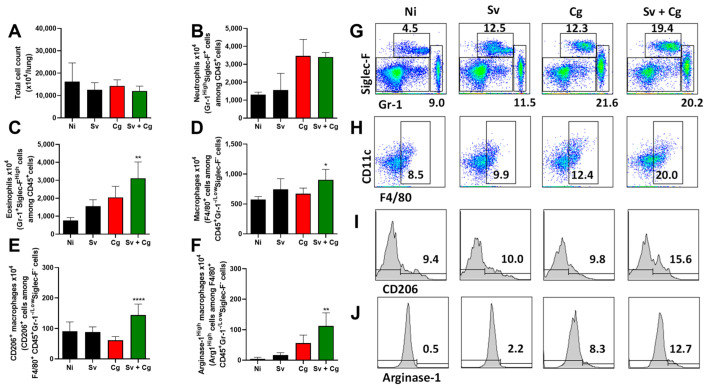
Cell recruitment to the lungs of Sv-Cg-coinfected mice by flow cytometry. CD45-positive cells among singlets were stratified according to Siglec-F and Gr-1 staining. Siglec-F-High and Gr-1-positive cells were defined as eosinophils, and Siglec-F-Low/negative and Gr-1-high cells were defined as neutrophils. The remaining cells were stratified according to CD11c and F4/80 staining, and F4/8-positive and CD11c^+^ cells were defined as macrophages. Macrophages were further analyzed for CD206 and Arginase-1 staining. Total cell count (**A**). Neutrophil count (**B**). Eosinophil count (**C**). Macrophage count (**D**). Macrophages are positive for DC206 staining (**E**). Macrophages are positive for arginase staining (**F**). Stratification of CD45-positive cells according to Siglec-F and Gr-1 staining (**G**). Stratification of the remaining cells according to CD11c and F4/80 staining (**H**). CD206 expression by macrophages (**I**). Arginase-1 expression by macrophages (**J**). ANOVA followed by Tukey’s multiple comparison test: * *p* < 0.01, ** *p* < 0.001, **** *p* < 0.0001 show statistical differences compared to the Cg group.

**Figure 6 jof-09-00968-f006:**
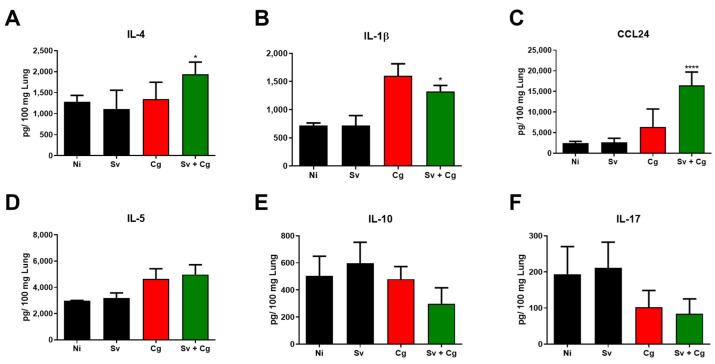
Cytokine levels in the lungs of Sv–Cg-coinfected mice. Concentrations of IL-4 (**A**), IL-1β (**B**), CCL24 (**C**), IL-5 (**D**), IL-10 (**E**), IL-17 (**F**), IFN-γ (**G**), TGF-β (**H**), TNF-α (**I**), and CXCL-2 (**J**) in the lungs of mice euthanized 10 days after infection with Cg. One-way ANOVA/Newman–Keuls multiple comparison tests: * *p* < 0.05, and **** *p* < 0.0001 represent the statistical difference compared to the Cg group at the same time point. NI: non-infected; Cg: group infected only with *C. gattii*; Sv: group infected only with *S. venezuelensis*; Sv + Cg: group infected with Sv 2 days before infection with *C. gattii*. The data are representative of two independent experiments.

## Data Availability

Not applicable.

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
