# Peer review of "The Th2 Response and Alternative Activation of Macrophages Triggered by Strongyloides venezuelensis Is Linked to Increased Morbidity and Mortality Due to Cryptococcosis in Mice"

_jof, 2023, doi:10.3390/jof9100968_

Round 1
Reviewer 1 Report
This study investigated the interaction between Strongyloides venezuelensis and Cryptococcus gattii in a murine model of coinfection. Overall the study is well designed and the manuscript is well-written. Thus, I just have only one minor suggestions.
1. please delete the sentence "Briefly, our results show that Sv infection increased mice mortality due to Cg by skewing host immune response to the fungus infection." (Line 75-77).
Author Response
Reviewer 1: This study investigated the interaction between Strongyloides venezuelensis and Cryptococcus gattii in a murine model of coinfection. Overall the study is well designed and the manuscript is well-written. Thus, I just have only one minor suggestions.
We really appreciate your comments. We performed the modification suggested in the text.
- please delete the sentence "Briefly, our results show that Sv infection increased mice mortality due to Cg by skewing host immune response to the fungus infection." (Line 75-77).
Response: Thank you for this comment. We deleted the sentence indicated.
Reviewer 2 Report
In this study, the authors examined how a nematode parasite infection affected the outcome of pulmonary infection of Cryptococci in a mouse model. They found that exposure of bone marrow derived macrophages with Strongyloides parasite antigens prior to incubation with Cryptococci reduced fungal phagocytosis and enhanced the intracellular survival of the yeast cells in an in vitro experiment, associated with decreased ROS production. Importantly, co-infection with Strogyloides parasites and Cryptococci in mice significantly reduced the survival of infected mice and enhanced the weight loss. This was accompanied with significantly enhanced fungal CFU in the lung and BALF, enhanced production of Arginase and nitric oxide, enhanced recruitment of eosinophils and CD206+ Arginase-1+ macrophages in the lung, and enhanced production of IL-4 and CCL24. The authors conclude that co-infection of nematode parasites in the lung promotes alternative activation of lung macrophages and Th2 responses, which facilitates the fungal growth in the lung. Overall, it is an interesting paper and the data are original.
Comments:
1) Since this manuscript addresses how parasite-induced alternative activation of macrophages affect fungal growth in the lung, the authors should add a paragraph either in the introduction section or in the discussion section to discuss how alternatively activation of lung macrophages in the lung affect fungal clearance and growth based on previous findings in the field. This is important because this background information is fundamental for performing the current experiment. Multiple groups in the field have made original contributions to the effect of alternative activation of lung macrophages on fungal growth including Hardison et al., AJP 176:774 (PMID: 20056835), Muller et al., J. Immunol 179:5637 (PMID: 17911623), Stenzel et al., AJP 174:486 (PMID: 19147811), and Strickland et al., iScience 26:106717 (PMID: 37216116). The findings of these publications are important and should be discussed in terms of alternative activation of lung macrophages.
2) For the title “Th2 response triggered….”, I believe that “The Th2 response and alternative activation of macrophages triggered by….” Is more accurately reflects the study.
3) There are many grammatical errors through the whole manuscript. For example: line 3, the title “mortality due cryptococcosis” should be “due to cryptococcosis” or “of cryptococcosis”; Line 53 “being S. stercoralis the main” should be “with S. stercoralis being the main…”; Line 93” mice response” should be “mouse response”; Line “mice Cg” should be “mouse Cg”; Line “mice immune” should be “mouse immune”; Line 345 “to the better characterize” should be “to better characterize”; Line 418, 419, 427 “mice” should be “mouse” etc…….
Moderate editing of English is required.
Author Response
Reviewer 2: In this study, the authors examined how a nematode parasite infection affected the outcome of pulmonary infection of Cryptococci in a mouse model. They found that exposure of bone marrow derived macrophages with Strongyloides parasite antigens prior to incubation with Cryptococci reduced fungal phagocytosis and enhanced the intracellular survival of the yeast cells in an in vitro experiment, associated with decreased ROS production. Importantly, co-infection with Strogyloides parasites and Cryptococci in mice significantly reduced the survival of infected mice and enhanced the weight loss. This was accompanied with significantly enhanced fungal CFU in the lung and BALF, enhanced production of Arginase and nitric oxide, enhanced recruitment of eosinophils and CD206+ Arginase-1+ macrophages in the lung, and enhanced production of IL-4 and CCL24. The authors conclude that co-infection of nematode parasites in the lung promotes alternative activation of lung macrophages and Th2 responses, which facilitates the fungal growth in the lung. Overall, it is an interesting paper and the data are original.
We really appreciate your comments. We have made the modifications suggested.
Comments:
- Since this manuscript addresses how parasite-induced alternative activation of macrophages affect fungal growth in the lung, the authors should add a paragraph either in the introduction section or in the discussion section to discuss how alternatively activation of lung macrophages in the lung affect fungal clearance and growth based on previous findings in the field. This is important because this background information is fundamental for performing the current experiment. Multiple groups in the field have made original contributions to the effect of alternative activation of lung macrophages on fungal growth including Hardison et al., AJP 176:774 (PMID: 20056835), Muller et al., J. Immunol 179:5637 (PMID: 17911623), Stenzel et al., AJP 174:486 (PMID: 19147811), and Strickland et al., iScience 26:106717 (PMID: 37216116). The findings of these publications are important and should be discussed in terms of alternative activation of lung macrophages.
Response: Thank you for this comment. We added a paragraph discussing how alternatively activation of lung macrophages affect fungal clearance and growth.
- For the title “Th2 response triggered….”, I believe that “The Th2 response and alternative activation of macrophages triggered by….” Is more accurately reflects the study.
Response: Thank you for this comment. We changed the title of the article.
3. There are many grammatical errors through the whole manuscript. For example: line 3, the title “mortality due cryptococcosis” should be “due to cryptococcosis” or “of cryptococcosis”; Line 53 “being S. stercoralis the main” should be “with S. stercoralis being the main…”; Line 93” mice response” should be “mouse response”; Line “mice Cg” should be “mouse Cg”; Line “mice immune” should be “mouse immune”; Line 345 “to the better characterize” should be “to better characterize”; Line 418, 419, 427 “mice” should be “mouse” etc…….
Response: Thank you for this comment. We corrected.
Reviewer 3 Report
Ludmila et al e aimed to investigate the interaction between Strongyloides venezuelensis (Sv) and Cryptococcus gattii (Cg) in a murine model of coinfection and in vitro. Their data points that primary Sv infection drives mice response through a Th2 polarization and compromises host’s response to Cg. In general, the results are impressive. The following issues needs to be addressed:
1. Prior to Cg infection, macrophages were stimulated with L3 antigens (10 µg/ml) for 131 3h. please clarify why 3h is chosen.
2. In the invivo study, what is the rationale that two distinct protocols were used?
3. “Cell viability was not affected by the exposure to the L3 antigens (Data not 230 shown).”-please provide this data as supplementary file.
4. For the phagocytosis, were these macrophages pretreated with LPS or IFNg?Naïve macrophages are always poor to engulf cryptococcus unless activated with LPS or IFNg.
5. In the results part, please use scatter plot to illustrate instead of column only. In addition, the number of experiments performed should be presented.
6. In fig 6B, the unit of y axis is missing.
Author Response
Reviewer 3: Ludmila et al e aimed to investigate the interaction between Strongyloides venezuelensis (Sv) and Cryptococcus gattii (Cg) in a murine model of coinfection and in vitro. Their data points that primary Sv infection drives mice response through a Th2 polarization and compromises host’s response to Cg. In general, the results are impressive. The following issues needs to be addressed:
- Prior to Cg infection, macrophages were stimulated with L3 antigens (10 µg/ml) for 131 3h. please clarify why 3h is chosen.
Response: Thank you for this comment. In a co-infection study of Influenza A virus and Cryptococcus gattii, murine macrophages were infected with the virus 2 hours before infection with the fungus. However, as there was no standardized stimulation time protocol for the L3 antigen and fungi, we decided to expose macrophages to L3 antigen 3 and 24 hours before Cg infection, and no significant results were observed when macrophages were exposed to antigen 24 hours before Cg infection. For this reason, the time of 3 hours was chosen to carry out the subsequent tests.
Oliveira, L. V. N., Costa, M. C., Magalhães, T. F. F., Bastos, R. W., Santos, P. C., Carneiro, H. C. S., Ribeiro, N. Q., Ferreira, G. F., Ribeiro, L. S., Gonçalves, A. P. F., Fagundes, C. T., Pascoal-Xavier, M. A., Djordjevic, J. T., Sorrell, T. C., Souza, D. G., Machado, A. M. V., & Santos, D. A. (2017). Influenza A Virus as a Predisposing Factor for Cryptococcosis. Frontiers in cellular and infection microbiology, 7, 419. https://doi.org/10.3389/fcimb.2017.00419
- In the in vivo study, what is the rationale that two distinct protocols were used?
Response: Thank you for this comment. We’ve used two protocols to standardize the best time for co-infecting the animals. First, we infected animals with Sv 7 days before or after to Cg infection. Considering that there was no difference in survival between the group infected with Sv before Cg and the group infected only with Cg, and that the infective larvae of Sv pass through the lungs about 2 days after infection with the nematode, we hypothesized that this time of co-infection with Sv before Cg could provide us a different result in terms of host response to coinfection. As we observed a lower survival in the group infected with Sv 2 days before infection with Cg, this time was chosen for further analyses.
- “Cell viability was not affected by the exposure to the L3 antigens (Data not 230 shown).”-please provide this data as supplementary file.
Response: Thank you for this comment. We will add the data as supplementary file.
- For the phagocytosis, were these macrophages pretreated with LPS or IFNg?Naïve macrophages are always poor to engulf cryptococcus unless activated with LPS or IFNg.
Response: Thank you for this comment. We did not pretreat the macrophages with LPS or interferon, considering that some groups would be exposed to the infective larval antigen of Strongyloides venezuelensis (L3), and the aim of our project was to evaluate whether this stimulus would be able to alter the phagocytosis of Cryptococcus by macrophages. For our work, we used the same methodology used in other works, including co-infection studies, where there is no pre-treatment of macrophages, but even so it is possible to obtain sufficient data for analysis and interpretation of the results obtained.
Freitas, G.J.C.; Gouveia-Eufrasio, L.; Emidio, E.C.P.; Carneiro, H.C.S.; de Matos Baltazar, L.; Costa, M.C.; Frases, S.; de Sousa Araújo, G.R.; Paixão, T.A.; Sossai, B.G.; et al. The Dynamics of Cryptococcus neoformans Cell and Transcriptional Remodeling during Infection. Cells 2022, 11, 3896. https://doi.org/10.3390/cells11233896
Peres-Emidio, E.C.; Freitas, G.J.C.; Costa, M.C.; Gouveia-Eufrasio, L.; Silva, L.M. v.; Santos, A.P.N.; Carmo, P.H.F.; Brito, C.B.; Arifa, R.D.N.; Bastos, R.W.; et al. Pseudomonas Aeruginosa Infection Modulates the Immune Response and Increases Mice Resistance to Cryptococcus Gattii. Front Cell Infect Microbiol 2022, 12, https://doi:10.3389/fcimb.2022.811474.
Peres-Emidio, E. C., Singulani, J. L., Freitas, G. J. C., Costa, M. C., Gouveia-Eufrasio, L., Carmo, P. H. F., Pedroso, S. H. S. P., Brito, C. B., Bastos, R. W., Ribeiro, N. Q., Oliveira, L. V. N., Silva, M. F., Paixão, T. A., Souza, D. D. G., & Santos, D. A. (2023). Staphylococcus aureus triggers a protective inflammatory response against secondary Cryptococcus gattii infection in a murine model. Microbes and infection, 25(6), 105122. https://doi.org/10.1016/j.micinf.2023.105122
- In the results part, please use scatter plot to illustrate instead of column only. In addition, the number of experiments performed should be presented.
Response: Thank you for this comment. In the manuscript, for formatting and presentation reasons we decided to keep only the bar graphs.
- In fig 6B, the unit of y axis is missing.
Response: Thank you for this comment. We corrected the figure.
Round 2
Reviewer 3 Report
The number of experiments (n=?) for each result should be shown in figure legends.
Author Response
Reviewer 3: The number of experiments (n=?) for each result should be shown in figure legends.
Response: Thank you for this comment. We added this information in the figure legends.